# Knowledge increases informative reporting by the public about urban coyotes

Abby Keller[1]*, Carly Sponarski[2], Chrystal Coleman[3], Colleen Cassady St. Clair[1]

**1** Department of Biological Sciences, University of Alberta, Edmonton, Alberta, Canada, **2** Northern Forestry Research Centre, Canadian Forest Service, Edmonton, Alberta, Canada, **3** Environment & Climate Resilience, Urban Planning and Economy Department, Edmonton, Alberta, Canada

☉ These authors contributed equally to this work.

* alkeller@ualberta.ca

## Abstract

Many municipalities use information about human-wildlife interactions collected in citizen-provided reports to monitor conflicts and guide management actions. However, high volumes of reports that describe benign wildlife behaviour can reduce the efficiency with which officials address reports that require management interventions, a situation that has occurred in Edmonton, Canada. We used data from a survey of Edmonton residents (n = 5,926) that asked respondents to anticipate whether they would alert officials if they witnessed (a) a coyote exhibiting benign behaviour in a natural area and (b) conflict-prone behaviour near human dwellings. We used path models to explore the predictors for two response variables; the agreement with reporting a benign sighting and the difference between scores for agreement to report conflict-prone behaviour and a benign sighting, which we interpreted as more informed reporting. As predictor variables, we considered a set of demographic, situational, cognitive, and experiential factors measured in the survey. A greater tendency to report the benign sighting was associated with prior 311 calls, higher risk perceptions, having experienced less severe interactions with coyotes, and greater knowledge of the consequences of food conditioning in coyotes. A greater tendency to anticipate reporting a conflict-prone coyote was associated with lower risk perceptions and greater knowledge of the consequences of food conditioning in coyotes, which is frequently associated with conflict. Further, individuals with higher risk perceptions were more likely to have experienced more severe interactions with coyotes, which were sometimes associated with living on a greenspace. Our results suggest that education campaigns could help people recognize benign behavior and identify and mitigate potential conflicts with coyotes. Education could occur as part of report receipt by city staff and outreach could target areas where coyote interactions are more likely, such as in residential areas along greenspaces.

**Data availability statement:** All relevant data used in this study are available from Zenodo at DOI 10.5281/zenodo.12669218 (https://zenodo.org/records/12669219).

**Funding:** This research was supported by a Natural Sciences and Engineering Research Council (NSERC) Discovery Grant (RES0061126) awarded to CCSC and an Alberta Graduate Excellence Scholarship awarded to ALK. The funders had no role in study design, data collection or analysis, manuscript preparation, or the decision to publish. There was no additional external funding received for this study.

**Competing interests:** The authors have declared that no competing interests exist.

## Introduction

Human-wildlife interactions have increased globally, driven by rapid urbanization, encroachment of humans on wildlife habitat, and wildlife adaptation to human-modified environments [1–3]. Many municipalities collect citizen-provided reports about human-wildlife interactions to monitor conflict and inform wildlife management actions [4,5]. Reporting of wildlife may also reflect public perceptions of a given species [6,7], which can further inform education campaigns and management strategies. However, the receipt of large volumes of reports describing benign wildlife behaviour can strain limited resources and reduce the efficiency with which officials can address reports that describe conflict-prone behaviour requiring management intervention. Refining reporting systems and educating the public to emphasize informative reports about conflict-prone situations can help officials respond more quickly and proactively. These goals can be supported and strengthened by targeted education campaigns that are informed by knowledge of the factors that influence the types of human-wildlife interactions people report. However, only a few studies to date have explored the drivers of public reporting behaviour to guide education and management. For example, individuals who reported nuisance alligators (*Alligator mississippiensis*) in Florida were more likely to live near alligator habitat, have children or pets in the home, and have greater risk perceptions of and lower tolerance for alligators [8]. In Colorado, individuals who reported black bears (*Ursus americanus*) were more likely to be dissatisfied with black bear management and have experienced more encounters with black bears [9].

Public reporting of coyotes (*Canis latrans*) is increasingly prevalent in urban areas where both sightings of and conflicts with coyotes have risen in recent years [4,10,11]. As with reporting of alligators, situational factors that increase the odds of an individual encountering a coyote, such as living near coyote habitat, may also increase the likelihood that an individual makes a report. Conflicts with coyotes are often reported in and near urban greenspaces and parks [11,12] and open areas [13]. The time of year may also influence the likelihood of conflicts and reporting of coyote encounters. For example, conflicts occur more often during the summer pup-rearing season, though people may see coyotes more frequently during autumn when young coyotes disperse [13]. Conflicts are also more likely when coyotes are food-conditioned (i.e., associate people with food) [3,14], in poor health [15], or when pets are present [10,16]. One's perception of the risk of conflict with or injury from coyotes may also influence whether one believes it is necessary to report an encounter with a coyote [17]. In Edmonton, Alberta, Canada (hereafter the City), the City government maintains records of coyote sightings and interactions reported to 311, a free phone service and app that connects residents to various municipal services. City rangers use these reports to monitor and guide management actions ranging from signage placement to hazing of bold animals and lethal removal of aggressive coyotes. Reports of coyotes in Edmonton have increased in the last decade [13], and large volumes of reports describing benign coyote behaviour have made it difficult for City officials to efficiently address reports that describe conflict-prone coyotes or other situations that require management intervention.

While previous literature has explored how a variety of factors influence human-wildlife conflict and reporting of human-wildlife interactions, much of this work has employed analyses that do not allow for both indirect and direct relationships to be tested simultaneously. We sought that advantage in this study, in which we used data from a public survey of Edmonton residents to build two path models [18] to explore how demographic, situational, cognitive, and experiential factors directly and indirectly predict whether someone (a) anticipates reporting a benign sighting of a coyote and (b) is more likely to report a conflict-prone coyote relative to a benign sighting of a coyote to 311. Our goal was to use information about these relationships to identify areas of focus for an ongoing education campaign and encourage reporting of encounters with coyotes that require management action while reducing the volume of reports describing benign sightings. The results of our research will support the City of Edmonton's goals of addressing human-coyote conflicts more efficiently and proactively, and similar approaches may be applied to the management of human-wildlife conflict beyond Edmonton. In our path model, we expected that respondents would anticipate reporting a benign sighting of a coyote and be more likely to report a conflict-prone coyote relative to a benign sighting in relation to demographic factors that may affect whether someone lives near coyote habitat [19], situational factors that increase the likelihood of encountering a coyote [8], experiential factors such as the severity of past interactions with coyotes [9], previous experience reporting a coyote to 311, and cognitive factors such as knowledge of food conditioning in coyotes and perceptions of risks posed by coyotes [8,17].

## Methods

In spring 2022, we assisted the City of Edmonton in designing a questionnaire based on discussion with City rangers and a review of literature focused on public perceptions of coyotes and other predators (S1 Appendix A). Using questionnaires from relevant literature as a guide [e.g., 10,20,21], we developed the questionnaire items to collect information that would help the City of Edmonton identify areas of focus for a public education campaign about urban coyotes intended to reduce human-coyote conflict in the City. Several City employees helped us to refine the survey before finalizing the version that was released to the public. The final questionnaire consisted of 78 items that targeted experiences with coyotes (3 items), observations of coyote attractants (2 items), knowledge of coyotes that access anthropogenic food resources (7 items), beliefs about coyotes (7 items), risk perceptions (16 items), personal reactions to coyote scenarios (8 items per scenario in two scenarios), opinions concerning coyote management (6 items in each of two scenarios), awareness of and comfort with hazing bold coyotes (2 items), past reporting of coyotes to the City 311 service (1 item), and demographic variables (12 items). The City administered the questionnaire through the Edmonton Insights Community, a pool of volunteers composed of Edmonton residents over the age of 15 who complete questionnaires and participate in discussions related to various municipal issues. In addition, the City posted an open weblink on its website and associated social media page, and Edmonton Urban Coyote Project members shared it with current volunteers and community liaisons. The survey was open for three weeks, from April 25, 2022, to May 15, 2022. The City of Edmonton provided the research team with access to the data on May 17, 2022. The University of Alberta's Research Ethics Board 2 (REB 2) determined that our secondary use of the City of Edmonton's questionnaire data in this research did not require Research Ethics Board review and provided us with an exemption for the use of the data (Pro00143661).

We used responses to two hypothetical coyote scenarios, one deemed by the research team to exhibit benign behaviour by a coyote and the other to be indicative of impending human-coyote conflict, to assess predicted or anticipated 311 call behaviour. Scenario 1, which we termed *Anticipates reporting benign sighting*, read, "*Imagine you are walking alone along a trail in a park, greenspace, or River Valley in Edmonton during the day, and a coyote crosses the trail 15 m (one bus length) ahead of you and stops to look at you.* We used the response to Scenario 1 as the outcome variable in our first path model. Scenario 2 read, "*Now imagine you are out walking alone in your neighbourhood during the day and see a coyote in the alleyway approaching yards. You know from your community social media site that several others have seen a coyote recently in the same area.* For each scenario, respondents were asked about their agreement that they would "…notify the

City via the 311 phone line or app." which had responses (-2) = *Strongly disagree,* (-1) = *Somewhat disagree,* (0) = *Neither agree nor disagree,* (+1) = *Somewhat agree*, (+2) = *Strongly agree*, and *Unsure*. Unsure responses were coded as missing. To better assess the difference between those who tended to report benign sightings (Scenario 1) and those who tended to report conflict-prone coyotes (Scenario 2), we recoded these responses to 1 (*strongly disagree*) to 5 (*strongly agree*) and subtracted responses to Scenario 1 from responses to Scenario 2. We then used the difference in responses as the outcome variable for our second model and called this variable *anticipates reporting conflict over sighting.*

Based on the survey questions that we believed to be most actionable by managers and consistent with existing literature [8,9,22–25], we selected six predictor variables with which to build our path model. These variables were: The highest level of education attained by the respondent (*education*), whether an individual resided along a greenspace (*property on greenspace),* the respondent's knowledge of the consequences of food conditioning in coyotes (*knowledge of food conditioning),* the most severe interaction the respondent had had with a coyote in the past year (*coyote interaction severity),* the respondent's perceptions of risk posed by coyotes to their personal safety, pet's safety, children's safety, and the risk of zoonotic disease transmission (*risk perceptions),* and whether the individual had previously reported a coyote to 311 (*previously called 311*). We left out outdoor pet ownership because the relationship between pets and human-coyote conflict is already well-established [10,16,26,27].

*Education* was used to assess the highest education level that the respondent had attained. This item read, "What is the highest level of education you have completed?" to which respondents could answer *Elementary/grade school graduate; High school graduate; College/technical school graduate; University undergraduate degree; Post-graduate degree (e.g., Masters, PhD); Professional school graduate (e.g., medicine, dentistry, veterinary medicine, optometry); I prefer not to answer. I prefer not to answer* responses were coded as missing.

*Property on greenspace* served as a measure of an individual's proximity to coyote habitat, which may increase the likelihood of encountering a coyote. Property on a greenspace, ravine, or other natural area was targeted with the item "*Do you have a yard facing or back onto a park or natural area (e.g., ravine, river valley, utility corridor)?*" for which responses were *Yes, No, Don't know*, and *I prefer not to answer*. *Don't know* and *I prefer not to answer* responses were coded as missing.

*Knowledge of food conditioning* was assessed using seven Likert-like items: "Thinking of coyotes that regularly access human sources of food in urban areas, to what extent would you agree with each of the following statements?": *They are more likely to survive and reproduce, They lose their fear of people, They become dependent on human sources of food, They are more likely to carry diseases, including some that people can get, They are more likely to be aggressive towards people or pets, They are more likely to den nearby,* and *They are more likely to be killed by wildlife managers to protect the public.* We tallied the number of correct responses to create a single score for our *knowledge of food conditioning* variable.

*Risk perceptions* about coyotes regarding personal safety, children's safety, pet safety, and disease transmission were assessed on a 5-point Likert-type scale. These items were preceded by the prompt: "*Given the presence of coyotes in Edmonton, how do you feel about each of the following? I am concerned about…*". Items were: (a)...*my own personal health or safety;* (b)…*my children's health or safety*; (c)…*my pet's health or safety*; and (d)…*the spread of diseases carried by coyotes.* Responses were (-2) = *Strongly disagree,* (-1) = *Somewhat disagree,* (0) = *Neither agree nor disagree,* (1) = *Somewhat agree*, (2) = *Strongly agree*, and *Not applicable*. *Not applicable* responses to any items (e.g., those without pets (25%) or children (36%)) were coded as missing. To simplify the model, we calculated a composite score for risk beliefs. To verify that composite scores could be calculated, we assessed the internal consistency of the scale using confirmatory factor analysis and then Cronbach's alpha to test the validity of our composite risk variable. We assessed the fit of the CFA using the following goodness-of-fit indices: chi-squared ($\Delta\chi^2$, $\chi^2/df$), root mean square error of approximation (RMSEA, an acceptable fit is < 0.05), comparative fit index (CFI, an acceptable fit is > 0.90), and Tucker-Lewis index (TLI, an acceptable fit is > 0.90). If the CFA was acceptable, we calculated Cronbach's alpha ($\alpha > .60$) to double-verify the construct.

*Coyote interaction severity* was based on the most severe interaction an individual indicated from a list of options: "In the last 12 months, which kind of encounter(s) have you had with a coyote?". Responses were *I have had no encounter with a coyote in the past 12 months* (0), *I saw a coyote from a car or building* (1), *I saw a coyote when I was outside of a car or building from a distance of at least 50 metres (approximately 3 city bus lengths)* (2), *I saw a coyote when I was outside and it was closer than 50 metres* (3), *A coyote approached me while I was walking, jogging, or cycling* (4), *A coyote tried to bite me or my pet* (5), *A coyote bit or killed my pet* (6).

*Previously called 311,* whether a respondent had previously reported a coyote to 311, a civic call centre and app for various city services, was assessed using a single item, "Have you ever called 311 (the City of Edmonton) to report a coyote?". Responses were *Yes*, *No*, and *Not Sure. Not sure* responses were coded as missing.

Using each of the two response variables, we fit a path model that explored the direct and indirect effects of *education, property on greenspace, knowledge of food conditioning, risk perceptions, coyote interaction severity,* and *previously called 311* (Figs 1 and 2). In both models, we controlled for respondent age and gender by regressing all other variables in the models on respondent age and gender. The path models were computed with the WLSMV estimator and linear regression in Mplus version 8.8 [28], which handles missing data through pairwise deletion. This technique made it possible to use each answered item from each respondent in the analysis instead of omitting all of the respondent's answers from the dataset if they did not answer all items in the questionnaire. We assessed model fit locally by examining the residuals and globally using the following goodness-of-fit indices: chi-squared ($\Delta\chi2$, $\chi2/df$), root mean square error of approximation (RMSEA, an acceptable fit is < 0.05), standardized root mean residual (SRMR, an acceptable fit is < 0.05), comparative fit index (CFI, an acceptable fit is > 0.90), and Tucker-Lewis index (TLI, an acceptable fit is > 0.90). In contrast to previous literature that explores reporting of wildlife, using path models allowed us to model both indirect and direct relationships between variables simultaneously, potentially better capturing the complex interactions between factors that may influence an individual reporting a coyote and reducing the risk of Type I error.

## Results

The questionnaire received a total of 5,926 responses. Of these, 4,959 responses came from Edmonton Insight Community members, 800 from the open weblink, 162 from the City surveys webpage, and 5 from the City of Edmonton website. Most

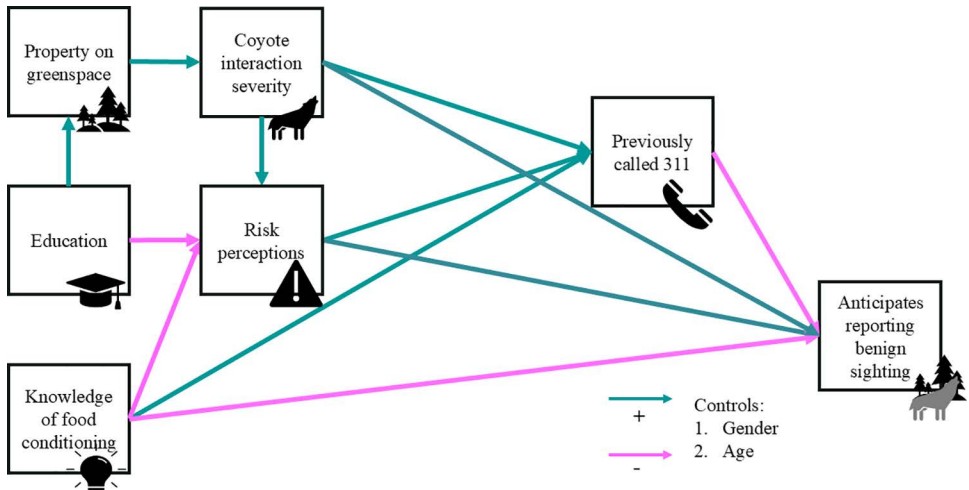

**Fig 1. Hypothesized path model showing the tendency to anticipate reporting a benign sighting of a coyote as a function of demographic, situational, cognitive, and experiential variables.** Paths hypothesized to be positive are shown in blue, and paths hypothesized to be negative are shown in pink.

respondents were over the age of 35 (85%). Over half were women (55%), did not identify as belonging to any marginalized or minority group (72%), and spoke English as the primary language in their household (93%). Complete descriptive information about the survey sample is detailed in Table 1, and complete survey results are provided in S1 Appendix A.

Most respondents anticipated reporting a conflict-prone coyote more than a benign sighting, but only slightly (mean difference in scores = 0.734, median = 0; Fig 3). About 25% of respondents agreed that they would report the benign sighting of a coyote (mean = -0.429 median = -1.00). Most respondents (82%) had completed at least some post-secondary education (Table 1), and about one-quarter (28.7%) lived near a greenspace, ravine, or other natural area (Table 1). Individuals were generally knowledgeable about the consequences of food conditioning in coyotes (mean = 4.23 median = 4.00). On average, survey respondents were neutral regarding risk beliefs about coyotes (Cronbach's alpha = 0.897, composite mean = -0.061). Few respondents (11.1%) had experienced a severe interaction with a coyote, though most had seen a coyote within the past year (mean = 2.149, median = 2.00). All risk items mapped well onto a single construct representing overall risk perceptions ($\chi^2(2)$ =154.13, RMSEA = 0.11, CFI = 0.986, TLI = 0.96). Eleven percent of respondents had previously reported a coyote to the City of Edmonton 311.

Our first model, predicting the reporting of a benign sighting, had good fit indices (Fig 4; $\chi^2(5)$ =58.193, RMSEA = 0.033, SRMR = 0.013, CFI = 0.979, TLI = 0.904), The modification indices did not recommend any logical changes to our paths. Most paths in the model were statistically significant, and calculating the $R^2$ values indicated that significant path components explained 0.8% to 24% of the variance. As we predicted, respondents with higher educational attainment were more likely to give on greenspaces and had lower risk perceptions about coyotes, property on a greenspace predicted more severe interactions with coyotes, and those who had experienced more severe interactions with coyotes tended to have higher risk perceptions. Contrary to our hypothesized model, those with more knowledge of the consequences of food conditioning in coyotes had higher risk perceptions. As we predicted, higher risk perceptions predicted a greater likelihood of anticipating reporting a benign sighting, but contrary to our expectations, severe coyote interactions decreased that likelihood. Knowledge of the consequences of food conditioning in coyotes and prior 311 calls increased the likelihood of someone anticipating reporting a benign sighting.

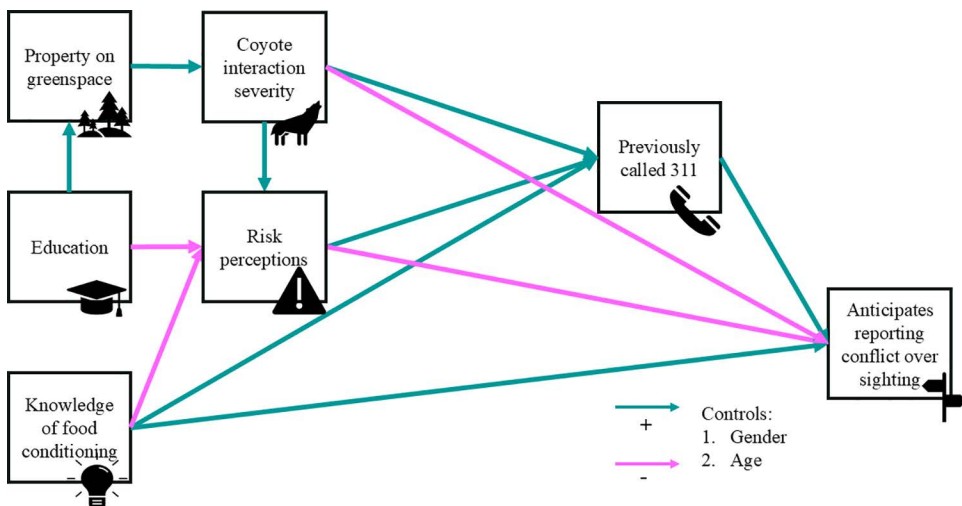

**Fig 2. Hypothesized path model showing the tendency to anticipate reporting a conflict-prone coyote relative to a benign sighting of a coyote as a function of demographic, situational, cognitive, and experiential variables.** This response variable subtracted agreement values for reporting a benign sighting from those for reporting a conflict-prone coyote. Paths hypothesized to be positive are shown in blue, and paths hypothesized to be negative are shown in pink.

**Table 1. Descriptive statistics for demographic information of questionnaire respondents.**

| Demographic Item | | Frequency | Percent |
|---|---|---|---|
| *Age* | | | |
| | 18–24 years old | 89 | 1.5 |
| | 25–34 years old | 683 | 11.5 |
| | 35–44 years old | 1210 | 20.4 |
| | 45–54 years old | 1085 | 18.3 |
| | 55–64 years old | 1264 | 21.3 |
| | 65 to 74 years old | 1096 | 18.5 |
| | 75+years old | 228 | 3.8 |
| | I prefer not to answer | 266 | 4.5 |
| | Under 18 years old | 5 | 0.1 |
| | | | |
| *Gender* | | | |
| | Man | 2165 | 36.5 |
| | Woman | 3270 | 55.2 |
| | Other | 81 | 1.4 |
| | I prefer not to answer | 410 | 6.9 |
| | | | |
| *Membership in marginalized or minority groups* | | | |
| | Racialized visible minority | 356 | 6.0 |
| | Persons with disabilities | 407 | 6.9 |
| | New to Canada | 48 | 0.8 |
| | None of these | 4263 | 71.9 |
| | Other | 193 | 3.3 |
| | I prefer not to answer | 649 | 11.0 |
| | | | |
| *Primary language spoken in household* | | | |
| | Arabic | 4 | 0.1 |
| | Cantonese | 17 | 0.3 |
| | English | 5501 | 92.8 |
| | French | 36 | 0.6 |
| | German | 8 | 0.1 |
| | Mandarin | 6 | 0.1 |
| | Other (Please specify) | 72 | 1.2 |
| | Punjabi | 10 | 0.2 |
| | Spanish | 17 | 0.3 |
| | Tagalog (Pilipino, Filipino) | 11 | 0.2 |
| | Ukrainian | 14 | 0.2 |
| | I prefer not to answer | 230 | 3.9 |
| | | | |
| *Education* | | | |
| | Elementary/grade school graduate | 28 | 0.5 |
| | High school graduate | 577 | 9.7 |
| | College/technical school graduate | 1596 | 26.9 |
| | University undergraduate degree | 1955 | 33.0 |

*(Continued)*

**Table 1.** (Continued)

| Demographic Item | | Frequency | Percent |
|---|---|---|---|
| | Post-graduate degree (e.g., Masters, PhD) | 1094 | 7.4 |
| | Professional school graduate (e.g., medicine, dentistry, veterinary medicine, optometry) | 239 | 4.0 |
| | I prefer not to answer | 437 | 7.4 |
| | | | |

Our second model, predicting greater agreement with reporting the conflict-prone coyote relative to the benign sighting, had good fit indices as well (Fig 5; $\chi^2(5)$ =35.224, RMSEA = 0.024, SRMR = 0.010, CFI = 0.982, TLI = 0.919), and the modification indices did not recommend any logical changes to our paths. Most of our path correlations were significant but generally had small effect sizes (Fig 5); calculating the $R^2$ values showed that significant correlations between path components explained between 0.09% and 12% of the variance, considerably less than the first model. In the second model, six of the relationships in the path were consistent with our predictions: property on a greenspace increased with education and predicted more severe interactions with coyotes; severe interactions increased the likelihood of previously calling 311; more education reduced risk perceptions of coyotes; and the tendency to report a conflict relative to the benign sighting increased with lower risk perceptions and greater knowledge of food conditioning. Three of our predicted relationships were not significant in the final path model: knowledge of food conditioning did not increase the likelihood of having called 311, and neither interaction severity nor prior calls to 311 predicted a higher tendency to report conflict over benign sightings. Contrary to our predictions, risk perceptions increased with knowledge of food conditioning.

## Discussion

Cities increasingly use citizen-provided reports of human-coyote interactions, but high volumes of reports that describe benign sightings of coyotes can increase the time spent processing all calls received and slow responses to reports that require management intervention. Using survey data from the City of Edmonton that described two scenarios about observing coyotes, we built two path models to test the potential direct and indirect causal effects of demographic, situational, cognitive, and experiential factors on an individual's tendency to report (a) a benign sighting of a coyote and (b) a conflict-prone coyote relative to a benign sighting of a coyote. We found moderate support for our predicted paths, with both models revealing that risk perceptions declined with education but increased with knowledge of food conditioning and predicted a greater likelihood of previous reporting. Further, previous reporting increased with the severity of past coyote interactions, which increased if respondents lived in a property abutting a greenspace, where residents tended to have higher educational attainment. Our first model revealed that anticipating reporting a benign sighting increased with previous reporting, higher risk perceptions, and greater knowledge of the consequences of food conditioning, but decreased with more severe prior interactions with coyotes. By contrast, our second model revealed that the tendency to report conflict-prone coyote relative to a benign sighting of a coyote decreased with risk perceptions and increased with knowledge of food conditioning.

We found a positive association between proximity to greenspaces and more severe interactions with coyotes. This result was similar to patterns observed in California, where reported human-coyote conflicts disproportionately occurred in parks [11], Calgary, Alberta, where conflicts often occurred in small greenspaces and nearer to a river [12], and Colorado, where conflicts more commonly occurred in open and developing areas [16]. Coyotes in Edmonton are also known to access diverse anthropogenic resources along the ecotone between natural and residential areas [29], and exposure to these resources can contribute to habituation, food conditioning, poor health, and, ultimately, conflict-prone behaviour toward humans [14,30]. Coyotes may also den in these ecotones, even incorporating anthropogenic materials into the structure of their dens [31]. Human activity in greenspaces increases during the spring and summer months when coyotes are raising their pups, and conflicts are known to be more common during the pup-rearing season [11,13,30].

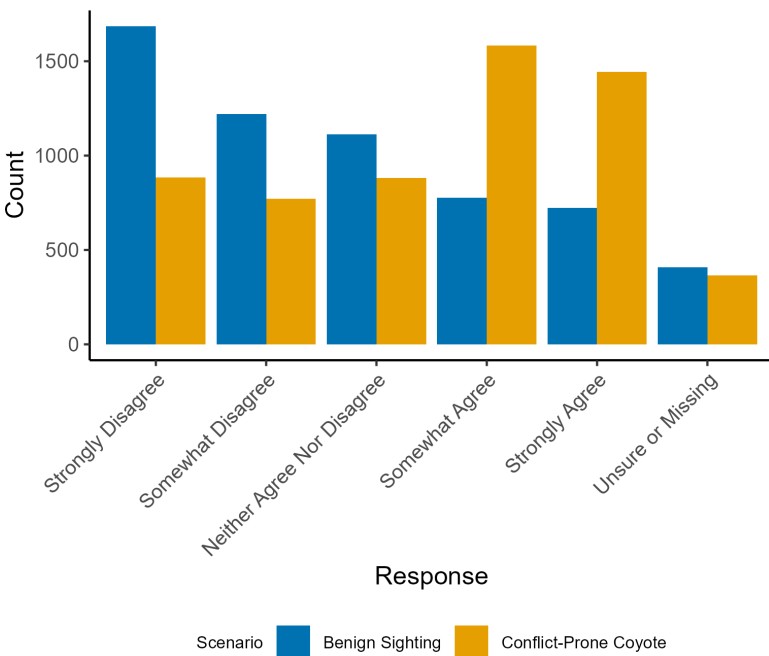

**Fig 3. Distribution of responses to the two coyote scenarios (benign and conflict-prone behaviours) presented in the questionnaire.**

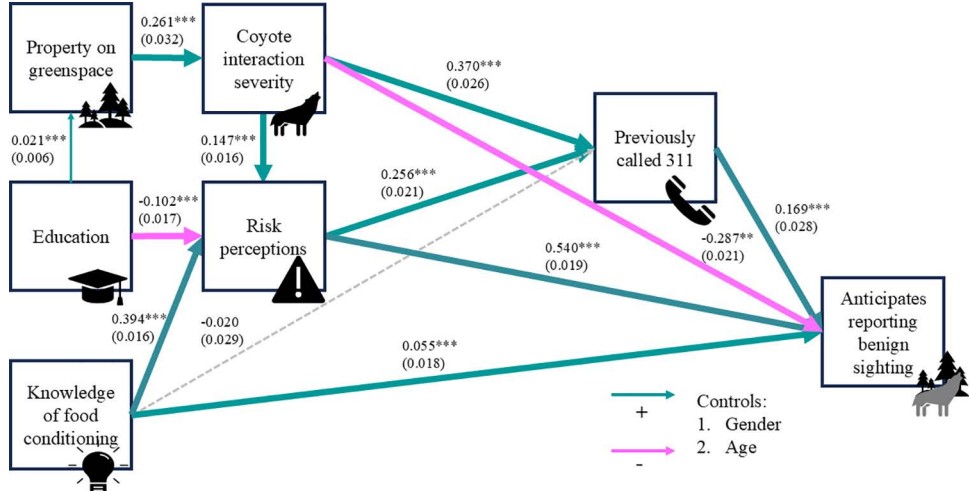

**Fig 4. Final path model showing tendency to report a benign sighting of a coyote as a function of demographic, situational, cognitive, and experiential variables.** Parameter estimates are displayed above their corresponding path with standard errors in parentheses. * indicates a path was significant at $p \leq 0.05$, ** for $p \leq 0.01$, and *** indicates that a path was significant at $p \leq 0.001$. Thicker paths indicate stronger causal relationships. Positive causal relationships are shown in blue, and negative relationships are shown in pink. Dotted lines denote paths that were not significant at $p \leq 0.05$.

The positive relationship between knowledge of food conditioning and risk perceptions differed from our expectations, as did the positive relationship between increased knowledge of food conditioning and a greater tendency to report a benign sighting. However, in our second model we found support for our hypotheses that knowledge of food conditioning increased the tendency to report conflict, whereas higher risk perceptions reduced that tendency. Previous research

has shown that knowledge can increase one's sense of agency over a situation and reduce risk perceptions [32], but we found that risk perceptions increased with knowledge of food conditioning in both models. These apparently contradictory results may indicate two things: in our hypothesized models, we assumed that risk perceptions may approximate one's fear of a species [8,17,33], which is typically associated with less tolerance [17] and a greater likelihood of reporting the species when it is encountered [8]. Alternatively, our measures of risk perceptions may also reflect informed awareness by respondents of the risks posed by coyotes, which are strongly associated with food conditioning [12,14,30]. This awareness may also explain why respondents tended to be more concerned about the safety of children or pets than about their own personal safety (S1 Appendix A). Although attacks on humans are rare, children are more commonly the victims of predatory attacks by coyotes [30]. Pets, meanwhile, are commonly involved in conflicts with coyotes [26,27], and areas with concentrations of pets and humans tend to be conflict hotspots [34]. In addition to the risk of physical conflict, coyotes pose a risk of transmitting diseases to people and pets [35,36]. Messaging about diseases such as the tapeworm parasite *Echinococcus multilocularis* has increased in Edmonton [36], possibly bolstering public awareness of this risk and increasing perceptions of risk associated with coyotes.

As we expected, those who had experienced a more severe interaction with a coyote in the past year were more likely to have higher risk perceptions and to have previously reported a coyote to the City of Edmonton 311 service. Interestingly, our first model showed that those who had experienced more severe interactions with coyotes were less likely to report a benign sighting, while those who had previously called 311 were more likely to report a benign sighting of a coyote. Given that experiencing a more severe interaction also predicted a greater likelihood of having previously reported to 311, this result may show that the experience of reporting to 311 could influence the types of interactions an individual anticipates reporting in future. However, in our second model, neither the severity of previous coyote interactions nor previous experience reporting to 311 significantly influenced whether an individual anticipated reporting a conflict-prone coyote relative to a benign sighting. Previous research from Colorado showed that those who had experienced more frequent interactions with black bears were likelier to report them [9]. If the same applies to human-coyote interactions, individuals

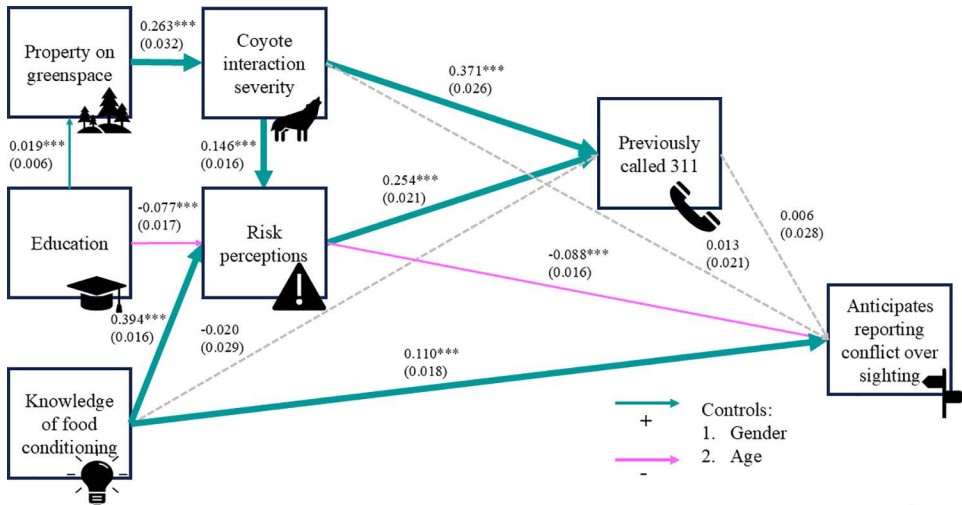

**Fig 5. Final path model showing tendency to report a conflict-prone coyote relative to a benign sighting as a function of demographic, situational, cognitive, and experiential variables.** This response variable subtracted agreement values for reporting a benign sighting from those reporting a conflict-prone coyote. Parameter estimates are displayed above their corresponding path with standard errors in parentheses. * indicates a path was significant at p ≤ 0.05, ** for p ≤ 0.01, and *** indicates that a path was significant at p ≤ 0.001. Thicker paths indicate stronger causal relationships. Positive causal relationships are shown in blue, and negative relationships are shown in pink. Dotted lines denote paths that were not significant at p ≤ 0.05.

                                                                      

who live near coyote habitat and see coyotes more frequently may become sensitized rather than habituating to their presence, leading them to become more likely to report benign sightings. Taken together, these results may indicate that 311 operators should provide clear information about risk factors and indicate which interactions warrant reporting.

There were some limitations to our survey and analytical approach that reduced the inferences our study can support. Importantly, the scenario we deemed benign, where a coyote stopped on a path 15 m away, would be much closer than many people have been to a coyote, even with the high rates of coyote observations our respondents described. This may have explained some of our unexpected results, reduced the difference in conflict potential people perceived between the two scenarios, and contributed to the generally small effect sizes we detected as predictors of our response variable for reporting conflict relative to benign scenarios. Although relatively small effect sizes are not unexpected given the often complex, multifaceted drivers of psychological phenomena and potential measurement error when using questionnaire data [37,38], they reduce the clarity of results for management purposes [39]. A second limitation is that the hypothetical responses to survey scenarios by individuals often overestimate their actual actions [40,41], which may also have reduced the difference in responses between the two scenarios. Third, although our sample was large, it was biased toward English speakers, women, older adults, people with some secondary education, and pet owners, which is not representative of all Edmontonians. The same demographic features that encourage survey responses may have encouraged more proactive reporting of coyotes in our scenarios as another way that the differences between them were minimized. Finally, this study is limited to urban coyote management in Edmonton, Alberta. Reporting and management needs may differ in other municipalities when managing human-wildlife interactions with coyotes and other species. Importantly, public perceptions and tolerance for wildlife differs among species, contexts, and locations [6,7,21,42], while the factors that influence reporting and other aspects of human-wildlife interactions are diverse, limiting the generalizability of the results of our model in other contexts.

## Management implications

Our results suggest that educational messaging to increase the frequency of reporting conflict-prone relative to benign observations of coyotes should be targeted in areas and seasons where coyotes are more frequently encountered, such as in residential areas adjacent to natural areas and during pup-rearing, when conflicts are more likely to occur. Educational messaging should promote knowledge of the conflict associated with food conditioning in coyotes and support realistic assessment of the risks posed by coyotes. This messaging should teach people to distinguish benign behaviour by coyotes in natural areas from food-seeking, conflict-prone behaviour in residential areas, where it might be mitigated with aversive conditioning [24,43,44]. Lastly, 311 operators may be uniquely positioned to provide education when they receive a report by helping people identify conflict-prone behaviour and directing individuals to additional sources of information. While our study focused on coyote reporting to a civic call centre, similar practices could be applied to other entities that receive reports about wildlife species in conflict with humans, including bears (*Ursus spp.*) [33,45,46], elk (*Cervus canadensis*) [47], and leopards (*Panthera pardus*) [48]. Because reporting factors are likely to differ among species, contexts, and jurisdictions, path models may provide a useful tool for investigating information contained in survey responses to inform subsequent education and management. Future research could identify and evaluate other approaches for increasing the information contained in public reports of wildlife and evaluate the effectiveness of education campaigns as tools for increasing human-wildlife coexistence in cities.

## Supporting information

**S1 Appendix A. Complete coyote survey results.** The Edmonton Coyote Insights Survey was administered online by the City of Edmonton in Spring 2022. The survey received a total of 5,926 responses.
(PDF)

## Acknowledgments

We respectfully acknowledge that our work was conducted on Treaty 6 Territory, which is a traditional gathering place for many Indigenous peoples, including the Anishinaabe/Ojibway/Salteaux, Blackfoot, Cree, Dene, Inuit, Iroquois, Nakota Sioux, Métis, and others. We thank the City of Edmonton staff who helped develop and implement the survey: Shawn Beskowiney, Jennifer Bewick, Troy Courtoreille, Denise Dion, Vishal Dutt, Greg Komarniski, Ryan Smar, Gareth Villanueva, Doug Yaceyko, and Shan Yang. We are grateful to the thousands of Edmonton residents who took the time to respond to the survey, providing us with a wealth of useful data. We are also thankful to Dr. Matthew Johnson for helpful guidance and feedback on early versions of this manuscript.

## Author contributions

**Conceptualization:** Abby Keller, Carly Sponarski, Chrystal Coleman, Colleen Cassady St. Clair.

**Data curation:** Abby Keller, Chrystal Coleman.

**Formal analysis:** Abby Keller.

**Funding acquisition:** Colleen Cassady St. Clair.

**Investigation:** Abby Keller, Chrystal Coleman.

**Methodology:** Abby Keller, Carly Sponarski, Colleen Cassady St. Clair.

**Resources:** Chrystal Coleman.

**Supervision:** Carly Sponarski, Colleen Cassady St. Clair.

**Writing – original draft:** Abby Keller.

**Writing – review & editing:** Abby Keller, Carly Sponarski, Chrystal Coleman, Colleen Cassady St. Clair.

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
