## [Decision Letter · Decision Letter 0]

16 Oct 2024

PONE-D-24-28515Knowledge of risks associated with food-conditioned coyotes increases the likelihood of informative reports by the publicPLOS ONE

Dear Dr. Keller,

Thank you for submitting your manuscript to PLOS ONE. After careful consideration, we feel that it has merit but does not fully meet PLOS ONE’s publication criteria as it currently stands. Therefore, we invite you to submit a revised version of the manuscript that addresses the points raised during the review process.

We look forward to receiving your revised manuscript.

Kind regards,

Vanessa Carels

Staff Editor

PLOS ONE

**Journal Requirements:**

This research was supported by a Natural Sciences And Engineering Research Council (NSERC) Discovery Grant (RES0061126) awarded to CCSC. The funders had no role in study design, data collection or analysis, manuscript preparation, or the decision to publish. 

We respectfully acknowledge that our work was conducted on Treaty 6 Territory, which is a traditional gathering place for many Indigenous peoples, including the Anishinaabe/Ojibway/Salteaux, Blackfoot, Cree, Dene, Inuit, Iroquois, Nakota Sioux, Métis, and others. We thank the City of Edmonton staff who helped develop and implement the survey: Shawn Beskowiney, Jennifer Bewick, Troy Courtoreille, Denise Dion, Vishal Dutt, Greg Komarniski, Ryan Smar, Gareth Villanueva, Doug Yaceyko, and Shan Yang. We are grateful to the thousands of Edmonton residents who took the time to respond to the survey, providing us with a wealth of useful data. We are also thankful to Dr. Matthew Johnson for helpful guidance and feedback on early versions of this manuscript. Funding for this study was provided by an Alberta Graduate Excellence Scholarship to ALK, a Natural Science and Engineering Research Council of Canada Grant to CCSC (RGPIN-2023-04892), and the University of Alberta.

This research was supported by a Natural Sciences And Engineering Research Council (NSERC) Discovery Grant (RES0061126) awarded to CCSC. The funders had no role in study design, data collection or analysis, manuscript preparation, or the decision to publish.

4. Please note that your Data Availability Statement is currently missing the DOI/accession number of each dataset OR a direct link to access each database. If your manuscript is accepted for publication, you will be asked to provide these details on a very short timeline. We therefore suggest that you provide this information now, though we will not hold up the peer review process if you are unable.

6. Please amend either the title on the online submission form (via Edit Submission) or the title in the manuscript so that they are identical.

**Additional Editor Comments:**

Comments from PLOS Editorial Office: We note that one or more reviewers has recommended that you cite specific previously published works. As always, we recommend that you please review and evaluate the requested works to determine whether they are relevant and should be cited. It is not a requirement to cite these works. We appreciate your attention to this request.

Reviewers' comments:

Reviewer's Responses to Questions

**Comments to the Author**

1. Is the manuscript technically sound, and do the data support the conclusions?

Reviewer #1: Partly

Reviewer #2: Yes

Reviewer #3: Yes

Reviewer #4: Yes

Reviewer #5: Yes

2. Has the statistical analysis been performed appropriately and rigorously? 

Reviewer #1: Yes

Reviewer #2: Yes

Reviewer #3: Yes

Reviewer #4: Yes

Reviewer #5: Yes

3. Have the authors made all data underlying the findings in their manuscript fully available?

Reviewer #1: No

Reviewer #2: Yes

Reviewer #3: No

Reviewer #4: Yes

Reviewer #5: Yes

4. Is the manuscript presented in an intelligible fashion and written in standard English?

Reviewer #1: Yes

Reviewer #2: Yes

Reviewer #3: Yes

Reviewer #4: Yes

Reviewer #5: Yes

5. Review Comments to the Author

**Reviewer #1** : 1. The title of this paper needs revised

2. Abstract is not well written (IMRaD is a standard format for writing abstracts in an orderly manner ....such as i. Introduction, ii Methodology, iii Results and iv Discussions). A summary of the result is missing from this abstract

3. Some Keywords are not refined enough (e.g The human body posture estimation)

4. The introduction can be enriched by briefly describe the state-of-the-art in the title of this study and provide more recently related references to support foundation of this studies to better context for the current research. I will recommend citing recently published related papers;

a. Statistical Analysis of Stakeholders Perception on Adoption of AI/ML in Sustainable Agricultural Practices in Rural Development. In X. S. Yang, S. Sherratt, N. Dey, & A. (. Joshi (Ed.), Proceedings of Ninth International Congress on Information and Communication Technology. ICICT 2024 2024. Lecture Notes in Networks and Systems. 1003. Springer, Singapore. doi: https://doi.org/10.1007/978-981-97-3302-6_11

b. Crime Prediction Using Twitter Sentiments and Crime Data. Informatica, 48(6), 35–42. doi: https://doi.org/10.31449/inf.v48i6.4749

5. The problem that author is trying to address in this paper is not clear yet, (clearer motivation for study is required)

6. Add the contribution of this study and study organization towards end of Introduction Section

7. Table 1 title need to be revised

8. In 4-5- lines, authors should summarize the literature gaps identified from the review of the before the methodology section

9. Author should provide the mathematical illustration for the evaluation

10. Services of language expert are required

11. Image quality needs to be improved

12. Ensure that all Figures are properly labeled and referenced

13. Authors should include the future work of this study

**Reviewer #2: ** 1- The title needs to be simplified further.

2- The abstract did not indicate the criteria adopted in stating the efficiency of this proposal.

3- Add a paragraph indicating the structure of the research with all its sections. It is placed at the end of the introduction.

4- Explain the related works and the negatives they suffered from and in which this manuscript excels.

5- The work algorithm needs to be simplified and explained in direct and clear points.

6-Where are the conclusions and future work?

**Reviewer #3:**  Clarify "Unsure" Responses: It would be helpful to provide more details on how you handled "unsure" responses in the path model. This would add transparency and ensure readers fully understand the treatment of this data.

Expand Practical Applications: The discussion could benefit from more details on how the findings can be applied in real-world urban wildlife management, particularly for educational campaigns and improving reporting systems.

Explain Unsupported Hypotheses: The reasoning behind some of the unsupported hypotheses, especially the unexpected relationship between knowledge of food conditioning and risk perceptions, needs further clarification to help readers understand these results.

Small Effect Sizes: Although the small effect sizes are acknowledged, they may limit the practical significance of the findings. It might be useful to discuss this limitation in more depth.

Sample Bias: The sample is somewhat biased toward older, English-speaking respondents. While this is noted, it could impact how generalizable the results are, so it's worth discussing this a bit further.

Limited Generalizability: Since the study is focused on Edmonton, the findings may not directly apply to other cities with different urban dynamics or wildlife. This could be emphasized more clearly in the limitations.

**Reviewer #4: ** - Based on previous similar works, the authors identified the unsolved challenges and proposed a study to examine how they could less influence the problem-based learning process.

- The paper is well-structured and documented.

- An overview on related works, some from recent years, is presented. It contains studies from specific literature.

- The theoretical background is well explained and properly used.

Areas of Improvements

- The study has some limitations, which supposed to be identified by the authors and clearly mentioned.

- Corrections should be made related to the use of English language.

**Reviewer #5:**  Strengths:

1. Large sample size provides good statistical power

2. Use of path analysis allows examination of direct and indirect effects

3. Clear rationale and relevance for urban wildlife management

4. Acknowledgment of limitations

Suggestions for improvement:

1. Provide more details on the survey development process and pilot testing

2. Further discuss potential reasons for small effect sizes

3. Expand on management implications, particularly regarding education/communication strategies

4. Consider adding a figure showing the distribution of responses to the two reporting scenarios

Overall, this is a well-conducted study that provides useful insights for improving citizen reporting of urban coyote encounters. The findings have practical implications for wildlife managers and public education efforts.

6. PLOS authors have the option to publish the peer review history of their article (what does this mean? ). If published, this will include your full peer review and any attached files.

**Do you want your identity to be public for this peer review?** For information about this choice, including consent withdrawal, please see our Privacy Policy .

Reviewer #1: No

Reviewer #2: No

Reviewer #3: **Yes: ** RAOOF ALTAHER

Reviewer #4: No

Reviewer #5: **Yes: ** Mounica Achanta

---

## [Author Response · Author response to Decision Letter 1]

9 Dec 2024

21 November 2024

Dear Editor,

Thank you for the invitation to revise and resubmit the manuscript “Knowledge of risks associated with food-conditioned by coyotes increases the likelihood of informative reporting by the public”. We are grateful for the helpful feedback from you and several reviewers and have adjusted the manuscript according to these suggestions. We provide a summary of those changes below and detail the changes on the following pages.

Our main changes were to simplify the title, refine keywords, improve figure image quality, correct typos and unclear language, and ensure the document complies with PLOS ONE formatting guidelines. We also edited the abstract to improve clarity and ensure it follows the IMRaD framework, added details and clarified phrasing about some background information, the goal of our study, and how and why we employed path analysis, and explained how we handled missing data. We elaborated on the limitations of our study, the implications our limitations have on the interpretation and application of our results, and how our approach can be applied to other human-wildlife conflict management contexts. We further explained potential future work that could stem from this path analysis. Finally, we removed funding information from the Acknowledgements section and amended our funding statement.

At the end of this letter, we copied the reviewers’ comments. We followed each comment with a description of how we addressed it (in the attached docx version of this letter, this also includes a screenshot showing the changes that we made). Thanks for your attention to our manuscript; we look forward to your further news.

Sincerely,

Abby L. Keller

Reviewer #1:

1. The title of this paper needs revised

We simplified the title while retaining key information as follows:

2. Abstract is not well written (IMRaD is a standard format for writing abstracts in an orderly manner ...such as i. Introduction, ii Methodology, iii Results and iv Discussions). A summary of the result is missing from this abstract

Our previous abstract followed this order, but we adjusted its content to illustrate the key results of our path model more clearly.

3. Some Keywords are not refined enough (e.g The human body posture estimation)

We refined the keywords to increase their relevance.

4. The introduction can be enriched by briefly describe the state-of-the-art in the title of this study and provide more recently related references to support foundation of this studies to better context for the current research. I will recommend citing recently published related papers;

a. Statistical Analysis of Stakeholders Perception on Adoption of AI/ML in Sustainable Agricultural Practices in Rural Development. In X. S. Yang, S. Sherratt, N. Dey, & A. (. Joshi (Ed.), Proceedings of Ninth International Congress on Information and Communication Technology. ICICT 2024 2024. Lecture Notes in Networks and Systems. 1003. Springer, Singapore. doi: https://doi.org/10.1007/978-981-97-3302-6_11

b. Crime Prediction Using Twitter Sentiments and Crime Data. Informatica, 48(6), 35–42. doi: https://doi.org/10.31449/inf.v48i6.4749

We reviewed these suggestions carefully but did not see how we could incorporate them into this paper. We’ll keep these references for potential future work though.

5. The problem that author is trying to address in this paper is not clear yet, (clearer motivation for study is required)

We adjusted the introduction to more clearly outline the problem we are addressing.

6. Add the contribution of this study and study organization towards end of Introduction Section

We added more details about the contribution of our study to the literature and the structure of our approach.

7. Table 1 title need to be revised

We adjusted the title to increase clarity.

8. In 4-5- lines, authors should summarize the literature gaps identified from the review of the before the methodology section

We added some additional information explaining relevant gaps in the literature and how our approach contributes to the literature about human-wildlife conflict reporting.

9. Author should provide the mathematical illustration for the evaluation

We elaborated on our rationale for using path analysis and added a citation that provides more information about how such models are handled mathematically.

10. Services of language expert are required

We reviewed the English throughout the manuscript.

11. Image quality needs to be improved

We increased the resolution of images, ensuring each had a dpi of 300 and that the dimensions aligned with PLOS ONE’s figure formatting guidelines.

12. Ensure that all Figures are properly labeled and referenced

We reviewed and reined figure captions and labelling.

E.g.,

13. Authors should include the future work of this study

We added more information about how our approach could be applied in other contexts of human-wildlife conflict and elaborated on potential future work.

Reviewer #2:

1- The title needs to be simplified further.

We simplified the title while ensuring it remains informative.

2- The abstract did not indicate the criteria adopted in stating the efficiency of this proposal.

We aren’t sure what is meant by this comment, but we reassessed the adherence of our abstract to contain the conventional information (IMRaD) and edited it to improve clarity and flow.

3- Add a paragraph indicating the structure of the research with all its sections. It is placed at the end of the introduction

We adjusted some phrasing and more clearly illustrated the structure of our approach but did not add an additional paragraph, preferring to keep additional details about our approach in the methods section which is consistent with published examples in PLOS ONE.

4- Explain the related works and the negatives they suffered from and in which this manuscript excels.

We elaborated on the gaps in the literature and limitations of past papers while more clearly explaining how this manuscript addresses these gaps and limitations.

5- The work algorithm needs to be simplified and explained in direct and clear points.

We reviewed the methods for clarity and completeness and added several details.

6-Where are the conclusions and future work?

We increased discussion about our results and the implications of our results with respect to Edmonton’s coyote education campaign and other human-wildlife conflict management contexts and included more explicit details about potential future work.

Reviewer #3: Clarify "Unsure" Responses: It would be helpful to provide more details on how you handled "unsure" responses in the path model. This would add transparency and ensure readers fully understand the treatment of this data.

We added information to the methods to explain how the WLSMV estimator handles missing values via pairwise deletion.

Expand Practical Applications: The discussion could benefit from more details on how the findings can be applied in real-world urban wildlife management, particularly for educational campaigns and improving reporting systems.

We added additional detail in the Management implications section explaining how our approach could be tailored to inform education, refine reporting, and support management in other human-wildlife interaction contexts.

Explain Unsupported Hypotheses: The reasoning behind some of the unsupported hypotheses, especially the unexpected relationship between knowledge of food conditioning and risk perceptions, needs further clarification to help readers understand these results.

In the discussion, we further explained our unsupported hypotheses and why we may have seen these results.

Small Effect Sizes: Although the small effect sizes are acknowledged, they may limit the practical significance of the findings. It might be useful to discuss this limitation in more depth.

We adjusted phrasing to emphasize the small effect sizes we observed and offered some potential reasons for them.

Sample Bias: The sample is somewhat biased toward older, English-speaking respondents. While this is noted, it could impact how generalizable the results are, so it's worth discussing this a bit further.

We included more details about how the bias in our sample may affect the generalizability of our results.

Limited Generalizability: Since the study is focused on Edmonton, the findings may not directly apply to other cities with different urban dynamics or wildlife. This could be emphasized more clearly in the limitations.

Similarly to above, we better explained how our focus on Edmonton presents limitations for generalizing our results to other municipalities or species. We also included more explanation about how our approach, rather than our exact results, can be applied to human-wildlife conflict reporting and management in other locations and contexts.

Reviewer #4: - Based on previous similar works, the authors identified the unsolved challenges and proposed a study to examine how they could less influence the problem-based learning process.

- The paper is well-structured and documented.

- An overview on related works, some from recent years, is presented. It contains studies from specific literature.

- The theoretical background is well explained and properly used.

Thank you for the positive feedback.

Areas of Improvements

- The study has some limitations, which supposed to be identified by the authors and clearly mentioned.

We changed some phrasing to be more explicit about our limitations and their implications for interpreting our results.

- Corrections should be made related to the use of English language.

We reviewed the entire manuscript to identify grammatical errors and awkward passages and adjusted several sentences to increase clarity.

Reviewer #5:

Strengths:

1. Large sample size provides good statistical power

2. Use of path analysis allows examination of direct and indirect effects

3. Clear rationale and relevance for urban wildlife management

4. Acknowledgment of limitations

Thank you for these comments; we appreciate them.

Suggestions for improvement:

1. Provide more details on the survey development process and pilot testing

We added some details to the methods to explain how we developed the survey. It was reviewed by several individuals at the City of Edmonton, but there was no pilot version, which we now explain.

2. Further discuss potential reasons for small effect sizes

We further explained why we may have seen small effect sizes.

3. Expand on management implications, particularly regarding education/communication strategies

Thank you for this suggestion. We adjusted some phrasing to improve clarity and further explained how our results can be applied to Edmonton’s coyote education campaign and how our approach can be used in other locations and contexts.

4. Consider adding a figure showing the distribution of responses to the two reporting scenarios

We appreciated this suggestion and added a bar plot as Figure 2 showing the distribution of responses to the scenarios. The final path model results are now shown in Fig 3.

Overall, this is a well-conducted study that provides useful insights for improving citizen reporting of urban coyote encounters. The findings have practical implications for wildlife managers and public education efforts.

Thank you.

---

## [Decision Letter · Decision Letter 1]

16 Jan 2025

PONE-D-24-28515R1Knowledge increases informative reporting by the public about urban coyotesPLOS ONE

Dear Dr. Keller,

Thank you for submitting your manuscript to PLOS ONE. After careful consideration, we feel that it has merit but does not fully meet PLOS ONE’s publication criteria as it currently stands. Therefore, we invite you to submit a revised version of the manuscript that addresses the points raised during the review process.

 **We appreciate your patience with the review process for your original submission. When I took over as AE for your revised manuscript, I reviewed the manuscript myself and recruited another reviewer (Reviewer #6) with expertise in carnivore conservation at the interface of human activity. Below you will also find a review (Reviewer #4) provided by one of your previous reviewers who does not have expertise in your field -- you can ignore those comments, Below you will find a few comments for your consideration,**

We look forward to receiving your revised manuscript.

Kind regards,

Stephanie S. Romanach, Ph.D.

Academic Editor

PLOS ONE

**Journal Requirements:**

Reviewers' comments:

Reviewer's Responses to Questions

**Comments to the Author**

1. If the authors have adequately addressed your comments raised in a previous round of review and you feel that this manuscript is now acceptable for publication, you may indicate that here to bypass the “Comments to the Author” section, enter your conflict of interest statement in the “Confidential to Editor” section, and submit your "Accept" recommendation.

Reviewer #4: All comments have been addressed

Reviewer #6: (No Response)

2. Is the manuscript technically sound, and do the data support the conclusions?

Reviewer #4: Yes

Reviewer #6: Yes

3. Has the statistical analysis been performed appropriately and rigorously? 

Reviewer #4: I Don't Know

Reviewer #6: Yes

4. Have the authors made all data underlying the findings in their manuscript fully available?

Reviewer #4: Yes

Reviewer #6: Yes

5. Is the manuscript presented in an intelligible fashion and written in standard English?

Reviewer #4: Yes

Reviewer #6: Yes

6. Review Comments to the Author

**Reviewer #4: ** Authors have incorporated the mentioned changes as raised in earlier version. Overall, the paper is good, organised, well written and the content is significant for the field related to the problem-based learning approaches in computer science.

**Reviewer #6: ** This study uses a thorough social survey and complimentary statistical analysis to evaluate the effects of personal experience, knowledge, and demographic factors on the likelihood of efficient reporting of human-coyote conflict. This is a widely relevant topic as coyote populations are flourishing in urban areas across the continent, and community reporting of wildlife conflict is becoming increasingly common worldwide. The paper's methods are fully appropriate for the questions being asked, and I believe the statistical approach to be sound. I was impressed with the survey response rate.

The major statistical limitation is that the manner of calculating reporting effectiveness left very little variation in the model's primary response variable, leading to small effect sizes and somewhat tenuous interpretation of the outputs. Did the authors try and perform the same analysis on other versions of the response variable, like the raw score of likelihood to report a conflict sighting (without subtracting the benign score)? I am curious how this would change the results or if more resolution could be gained.

As mentioned by other reviewers, the embedded figures were low resolution but that may have been addressed with additional file uploads. The manuscript's writing/grammar showed no issues, and it is overall well-written and understandable.

Line comments:

275: Remove the word 'greater'

303-305: Was this result presented in the paper? If not, some support is needed

7. PLOS authors have the option to publish the peer review history of their article (what does this mean? ). If published, this will include your full peer review and any attached files.

**Do you want your identity to be public for this peer review?** For information about this choice, including consent withdrawal, please see our Privacy Policy .

Reviewer #4: No

Reviewer #6: No

---

## [Author Response · Author response to Decision Letter 2]

2 Apr 2025

Thank you for reviewing our manuscript. Below are responses to specific comments. These comments, along with screenshots showing the changes made to the manuscript, are also included in our Response to Reviewers letter.

Reviewer #4: Authors have incorporated the mentioned changes as raised in earlier version. Overall, the paper is good, organised, well written and the content is significant for the field related to the problem-based learning approaches in computer science.

Thank you.

Reviewer #6: This study uses a thorough social survey and complimentary statistical analysis to evaluate the effects of personal experience, knowledge, and demographic factors on the likelihood of efficient reporting of human-coyote conflict. This is a widely relevant topic as coyote populations are flourishing in urban areas across the continent, and community reporting of wildlife conflict is becoming increasingly common worldwide. The paper's methods are fully appropriate for the questions being asked, and I believe the statistical approach to be sound. I was impressed with the survey response rate.

Thank you.

The major statistical limitation is that the manner of calculating reporting effectiveness left very little variation in the model's primary response variable, leading to small effect sizes and somewhat tenuous interpretation of the outputs. Did the authors try and perform the same analysis on other versions of the response variable, like the raw score of likelihood to report a conflict sighting (without subtracting the benign score)? I am curious how this would change the results or if more resolution could be gained.

Thank you for this suggestion. Following this, we ran another version of the model with the raw score of likelihood to report a benign sighting as the response variable. We chose to use the benign sighting score rather than the conflict-prone score because we felt understanding what factors predict anticipated reporting of the benign scenario could better support the City of Edmonton’s goal of reducing the volume of reports received about benign sightings of coyotes. We placed this new model before our original model throughout the text for the sake of flow and clarity with respect to our original model.

The results of this model provided further support for and helped us to refine our recommendations for a coyote education campaign aimed at reducing reports of benign sightings and encouraging reports of conflict-prone coyotes.

As mentioned by other reviewers, the embedded figures were low resolution but that may have been addressed with additional file uploads. The manuscript's writing/grammar showed no issues, and it is overall well-written and understandable.

Line comments:

275: Remove the word 'greater'

Done.

303-305: Was this result presented in the paper? If not, some support is needed

Thanks for catching this. We did not explicitly present this in the results, but we added a reference to Appendix A, where the responses to the risk items are presented in detail.

---

## [Editor Report · Decision Letter 2]

10 Apr 2025

Knowledge increases informative reporting by the public about urban coyotes

PONE-D-24-28515R2

Dear Dr. Keller,

We’re pleased to inform you that your manuscript has been judged scientifically suitable for publication and will be formally accepted for publication once it meets all outstanding technical requirements.

Kind regards,

Stephanie S. Romanach, Ph.D.

Academic Editor

PLOS ONE

---

## [Editor Report · Acceptance letter]

PONE-D-24-28515R2

PLOS ONE

Dear Dr. Keller,

I'm pleased to inform you that your manuscript has been deemed suitable for publication in PLOS ONE. Congratulations! Your manuscript is now being handed over to our production team.

Kind regards,

on behalf of

Dr. Stephanie S. Romanach

Academic Editor

PLOS ONE